

# Composition dependence of the specific heat of FeSi

Carolina Burger[1⋆], Andreas Bauer[1,2] and Christian Pfleiderer[1,2,3]

1 Physik-Department, Technische Universität München, D-85748 Garching, Germany
2 Zentrum für Quantum Engineering (ZQE), Technische Universität München,
D-85748 Garching, Germany
3 Munich Center for Quantum Science and Technology (MCQST),
Technische Universität München, D-85748 Garching, Germany

⋆ carolina.burger@tum.de

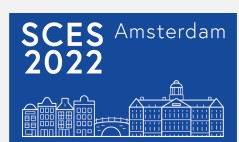

*International Conference on Strongly Correlated Electron Systems
(SCES 2022)
Amsterdam, 24-29 July 2022*

## Abstract

Recently, a high-mobility surface conduction channel and in-gap states were identified in the correlated small-gap semiconductor FeSi using electrical transport measurements and high-resolution tunneling spectroscopy. The mobility of the charge carriers in the surface channel is quantitatively reminiscent of topological insulators, but displays a lack of sensitivity to the presence of ferromagnetic impurities as studied by means of a series of single crystals with slightly different starting compositions. Here, we report measurements of the specific heat of these crystals. At low temperatures, a shallow maximum is observed in the specific heat divided by temperature. This maximum is suppressed under magnetic field, characteristic of a Schottky anomaly associated with magnetic impurities. In comparison, the height of this maximum decreases with increasing initial iron content.

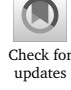

## 1 Introduction

FeSi is a correlated small-gap semiconductor in which an unusual temperature dependence of the electrical and magnetic properties has been attracting scientific interest for several decades [1–3]. As illustrated by means of the temperature dependence of the electrical resistivity shown in Fig. 1(a), FeSi exhibits a crossover around 200 K between a paramagnetic metal with strong spin fluctuations at high temperatures, denoted regime I, and a semiconducting state with reduced magnetic susceptibility featuring an energy gap of about 60 meV, denoted regime II [4–8]. For decreasing temperature, the resistivity continues to increase at a reduced slope below 100 K, denoted regime III, followed by a saturation on logarithmic scales

at low temperatures, denoted regime IV [9,10]. The magnetic susceptibility increases by about two orders of magnitude in regimes III and IV [11]. While band structure calculations established unambiguously that FeSi is a band insulator at low temperatures [12–14], the unusual metallization and paramagnetism at high temperatures was attributed to correlation-induced incoherence under increasing temperature [15]. The saturation of the resistivity at low temperatures was attributed to the emergence of an impurity band, with ferromagnetic impurities potentially adding to the complexity of the low-temperature properties [16–18].

Recently, the emergence of a high-mobility surface conduction channel at low temperatures was inferred from the electrical transport properties of a series of single crystals of FeSi prepared under systematic variation of the initial iron content using the optical floating-zone technique [19–23]. This observation was corroborated by means of measurements on thin needles grown from tin flux [24] as well as high-resolution tunneling spectroscopy that revealed two in-gap states in the low-temperature regime of the samples grown by the floating-zone technique [25]. The surface-to-bulk ratios of the charge carrier densities and mobilities observed in the transport properties compare quantitatively with values observed in topological insulators such as $Bi_2Te_3$ [19, 20, 26]. Most notably, the surface channel in FeSi appears to exhibit a remarkable robustness against the presence of ferromagnetic impurities. An open question concerns, in turn, whether this robustness represents a hallmark of FeSi that is also reflected in bulk properties.

## 2 Experimental Methods

In this paper, we report a study of the specific heat of the same series of single crystals studied in Refs. [19, 20], as prepared by means of the optical floating-zone technique using slightly different starting compositions $Fe_{1+x}Si$ [21–23]. The magnetization and electrical transport properties of these single crystals were reported in Refs. [19, 20, 25]. In addition, a single crystal with an iron deficiency $x = -0.005$ was studied. Samples cut from the start of the single-crystal growth process (close to the initial grain selection) and from the end (close to the final quenched zone) were investigated as summarized in Tab. 1. Consistent with the detection limits of standard techniques for metallurgical characterization, such as powder x-ray diffraction or energy-dispersive x-ray spectroscopy, and the tiny variation of the starting compositions, no systematic variations of the composition of the samples after the growth process were resolved using these methods. In comparison, studies of the density and nature of structural point defects using techniques such as positron annihilation spectroscopy, planned for the future, may provide valuable insights, as demonstrated on isostructural $Mn_{1+x}Si$ [27, 28]. Such studies, however, were well beyond the scope of the work presented in this manuscript.

For the present study, cubes with an edge length of 1 mm were prepared, each with two surfaces perpendicular to ⟨100⟩ and four surfaces perpendicular to ⟨110⟩. The specific heat measurements were carried out in a Quantum Design physical property measurement system at temperatures down to 1.9 K and under magnetic fields up to 14 T. The single crystal cubes were mounted on the platform of the measurement puck by means of a tiny amount of Apiezon N grease. Prior to mounting each sample, the heat capacity of the grease was measured in order to subtract it from the total heat capacity. Precise subtraction proved to be crucial for the determination of the heat capacity of the $Fe_{1+x}Si$ samples.[1] All measurements were carried out using a quasi-adiabatic large heat pulse technique, in which heat pulses had a size of 30% of the temperature at the start of the pulse [29]. For each specific heat curve, data were measured at 80 starting temperatures in a logarithmic spacing, covering the temperature regime from 1.9 K

---

[1]As part of the studies reported here, we noticed that the specific heat presented in the supplemental material of Ref. [25] is dominated by contributions of the grease and hence erroneous. An erratum will be published separately.

Table 1: Overview of the samples studied in this paper (see also Refs. [19, 20, 25]). For each sample, the chemical composition of the polycrystalline rods before float-zoning and the location from which the sample was cut within the float-zoned single-crystal ingot are stated.

| Sample | Starting composition | Location in float-zoned ingot |
|--------|---------------------|-------------------------------|
| A1 | $Fe_{0.99}Si$ | start |
| A2 | $Fe_{0.99}Si$ | end |
| AB | $Fe_{0.995}Si$ | start |
| B1 | FeSi | start |
| B2 | FeSi | end |
| C | $Fe_{1.01}Si$ | start |

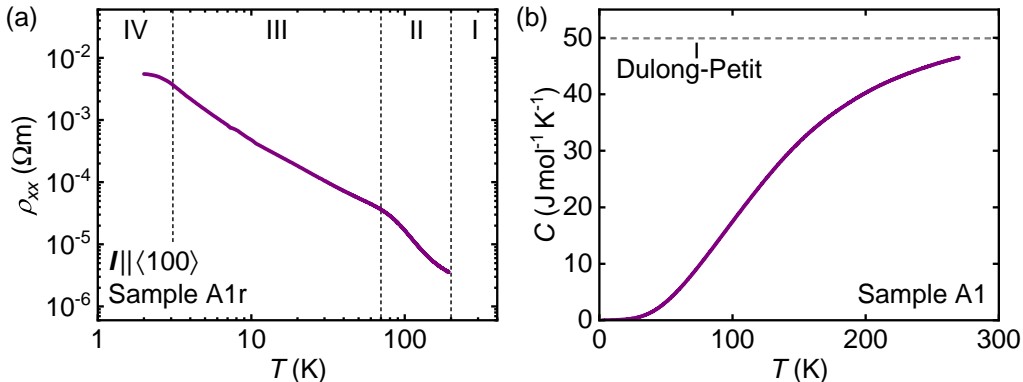

Figure 1: Temperature dependence of the low-temperature properties of FeSi for samples A1r and A1 in zero magnetic field. (a) Electrical resistivity for current along $\langle 100 \rangle$ on a double-logarithmic scale. Four regimes may be distinguished as a function of temperature, denoted I through IV; see text for details. Data taken from Refs. [19, 20]. (b) Specific heat. No anomalies suggestive of phase transitions are observed in the temperature range studied.

to 270 K. The heat pulses and concomitant data collection were repeated three times at each temperature.

## 3 Experimental Results

A typical temperature dependence of the specific heat of FeSi is shown in Fig. 1(b), for the case of sample A1. Both the resistivity shown in Fig. 1(a) and the specific heat shown in Fig. 1(b) were measured on samples cut from the same location of the same ingot, referred to as samples A1r and A1, respectively. The specific heat as a function of temperature is characteristic of a nonmagnetic crystal in which phonon contributions dominate. It approaches the Dulong–Petit value of $6R = 49.9 \, \text{J} \, \text{mol}^{-1} \text{K}^{-1}$ at high temperatures. No anomalies suggestive of phase transitions were observed in any of the samples in the temperature and field range investigated.

For the analysis of our data, we consider the specific heat divided by temperature, $C/T$, as illustrated for sample A1 in Fig. 2(a). A prominent feature concerns a shallow maximum below $\sim 10$ K, consistent with Ref. [16], where the maximum was attributed to a Schottky

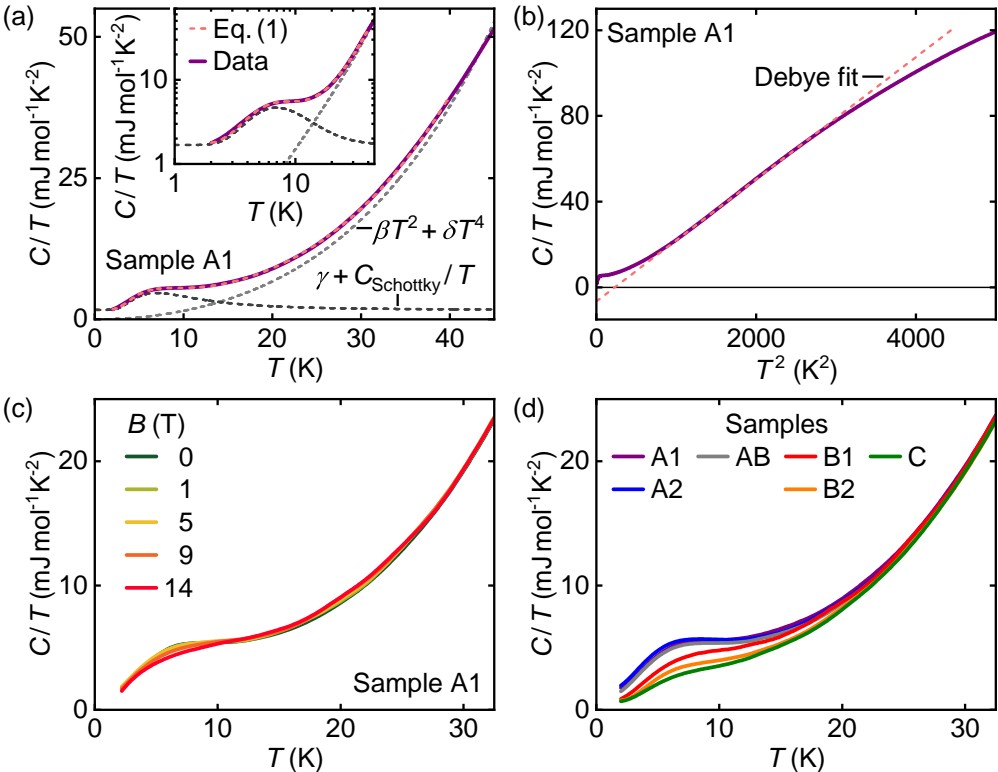

Figure 2: Specific heat divided by temperature as function of temperature. (a) Data of sample A1 in zero magnetic field. At low temperatures, a shallow maximum is observed that is suggestive of a Schottky anomaly. The dashed red line corresponds to a fit according to Eq. (1). The dashed dark and light gray lines correspond to the sum of the terms dominating at low and high temperatures, respectively. Inset: Data on a double-logarithmic scale. (b) Specific heat divided by temperature as a function of temperature squared. The dashed line represents a linear regression corresponding to a phonon contribution as expected in a Debye model. See text for details. (c) Specific heat divided by temperature of sample A1 under selected magnetic fields up to 14 T. With increasing field, the maximum suggestive of a Schottky anomaly is suppressed. (d) Zero-field data for samples with different initial compositions. At temperatures above ∼20 K, the data essentially track each other. At low temperatures, the height of the maximum decreases with increasing iron content in the starting composition prior to crystal growth.

anomaly. In Ref. [16], an additional maximum was reported in the specific heat at temperatures well below 2 K, i.e., below the temperature range investigated in our study. To fit our data we use the empirical description suggested in Ref. [16], however, taking into account a single Schottky anomaly only. Beyond the conventional terms proportional to $T$ and $T^3$, an additional contribution proportional to $T^5$ as well as the Schottky anomaly were included, resulting in

$$C = \gamma T + \beta T^3 + \delta T^5 + a_1 \frac{(T_1/T)^2 \exp(T_1/T)}{[1 + \exp(T_1/T)]^2} . \tag{1}$$

When fitting the coefficients $\gamma$, $\beta$, $\delta$, and $a_1$ as well as the Schottky temperature $T_1$ for sample A1, the following values are obtained, as summarized in Tab. 2: $\gamma_{A1} = 1.68$ mJ mol$^{-1}$K$^{-2}$, $\beta_{A1} = 1.45 \cdot 10^{-2}$ mJ mol$^{-1}$K$^{-4}$, $\delta_{A1} = 5.67 \cdot 10^{-6}$ mJ mol$^{-1}$K$^{-6}$, $a_{1,A1} = 54.3$ mJ mol$^{-1}$K$^{-1}$, and $T_{1,A1} = 22.5$ K. The corresponding fit is shown as a dashed red line in Fig. 2(a). The

value of $\beta_{A1}$ corresponds to a Debye temperature $\Theta_{D,A1} = 645$ K. It may be helpful to note that when fitting the data without the contribution proportional to $T^5$, values of $\gamma$ are negative and thus not physical. This observation is illustrated in Fig. 2(b) showing a linear fit of $C/T$ as a function of $T^2$, where the axis intercept corresponds to $\gamma$ and the slope to $\beta$. The values of $\beta$ inferred without the $T^5$ contribution translate to Debye temperatures of the order of 500 K, consistent with values reported for other isostructural transition-metal compounds for which the data were analyzed without $T^5$ terms as well [30, 31].

Integration of the term describing the Schottky anomaly yields an estimate for the underlying entropy. For sample A1, we obtain $\Delta S_{1,A1} = 37.3$ mJ mol$^{-1}$K$^{-1} \approx 0.006\,R\ln 2$, corresponding to about 0.006 two-level centers per formula unit of FeSi. As no data were measured below 2 K in our study, we cannot exclude the putative presence of a second Schottky anomaly at very low temperatures previously reported in Ref. [16]. Fitting our data, we estimate that such an anomaly may yield an entropy not larger than $\Delta S_{2,A1} < 0.002\,R\ln 2$. This concentration suggests that the two-level centers are located in the bulk of the material. For comparison, when assuming that the two-level centers emerge at the surface of the sample and that each formula unit may support a single two-level center, a surface layer of a thickness of $\sim 1$ $\mu$m would be required, i.e., much thicker than typically observed for surface-induced phenomena.

It is instructive to note that, in contrast to the specimens investigated in our study, the samples studied in Ref. [16] were grown from vapor transport. Various materials properties, such as the magnetization and the Hall effect reported in Ref. [16], as well as tests we performed ourselves on samples of FeSi grown from vapor transport consistently suggest that such samples may contain substantial concentrations of magnetic impurities, such as elemental iron. In turn, when comparing our results to those reported in Ref. [16], namely $\gamma_P = 1.1$ mJ mol$^{-1}$K$^{-2}$, $\beta_P = 0.91 \cdot 10^{-2}$ mJ mol$^{-1}$K$^{-4}$, $\delta_P = 11 \cdot 10^{-6}$ mJ mol$^{-1}$K$^{-6}$, $a_{1,P} = 9.2$ mJ mol$^{-1}$K$^{-1}$, $T_{1,P} = 6.8$ K, $a_{2,P} = 11$ mJ mol$^{-1}$K$^{-1}$, $T_{2,P} = 0.95$ K, $\Delta S_{1,P} = 6.3$ mJ mol$^{-1}$K$^{-1}$, and $\Delta S_{2,P} = 7.9$ mJ mol$^{-1}$K$^{-1}$, two key differences become apparent. First, compared to our results, $\beta$ is smaller by a factor of 1.6 while $\delta$ is larger by a factor of 2. In a fit using Eq. (1), these two parameters are connected, where smaller values of $\beta$ result in larger values of $\delta$ and vice versa. Since in Ref. [16] specific heat data were measured down to temperatures as low as 60 mK and presented on a double-logarithmic scale up to 35 K, the behavior at high temperatures may have been accounted for less accurately. Note that $\beta_P = 0.91 \cdot 10^{-2}$ mJ mol$^{-1}$K$^{-4}$ corresponds to a Debye temperature $\Theta_{D,P}^* = 753$ K instead of the value $\Theta_{D,P} = 377$ K stated in Ref. [16]. Second, the Schottky anomaly at $T_1$ is smaller and shifted to lower temperatures. As discussed below, this anomaly is sensitive to the detailed composition of the sample, where the values reported in Ref. [16] are consistent with a large iron content.

As illustrated in Fig. 2(c), under magnetic fields up to 14 T, the maximum at $T_1$ observed in our samples decreases in height and the associated entropy release shifts to higher temperatures. Such a field dependence suggests qualitatively that the maximum is linked to magnetic degrees of freedom, consistent, for instance, with magnetic impurities.

Comparing the specific heat of different samples, as shown in Fig. 2(d) in terms of $C/T$ in zero magnetic field, several characteristics appear to be the same for all compositions. First, at temperatures above $\sim 20$ K data for all samples studied track each other, indicating essentially identical contributions due to phonons at high temperatures. This finding suggests that the small variations of the starting composition do not affect the crystal structure on a fundamental level. Second, all samples exhibit a shallow maximum at low temperatures, suggestive of a Schottky anomaly as discussed above. The height of this anomaly varies systematically between samples. Third, for all samples studied, the specific heat is in excellent agreement with Eq. (1). The coefficients inferred from these fits are summarized in Tab. 2. Fourth, in all samples studied, the specific heat at high temperatures is insensitive to applied magnetic fields up to 14 T (not shown).

Table 2: Overview of key parameters inferred from the specific heat of FeSi for the samples investigated in our study. For each sample, the coefficients $\gamma$, $\beta$, and $\delta$ are shown together with the coefficient $a_1$ and the characteristic temperature $T_1$ describing the Schottky anomaly at low temperatures. In addition, the Debye temperature $\Theta_D$ calculated from the coefficient $\beta$ and an estimate of the entropy $\Delta S_1$ associated with the Schottky anomaly are presented.

| Sample | $\gamma$ [mJ mol$^{-1}$K$^{-2}$] | $\beta$ [$10^{-2}$ mJ mol$^{-1}$K$^{-4}$] | $\delta$ [$10^{-6}$mJ mol$^{-1}$K$^{-6}$] | $a_1$ [mJ mol$^{-1}$K$^{-1}$] | $T_1$ [K] | $\Theta_D$ [K] | $\Delta S_1$ [mJ mol$^{-1}$K$^{-1}$] |
|--------|------|------|------|------|------|------|------|
| A1 | 1.68 | 1.45 | 5.67 | 54.3 | 22.5 | 645 | 37.3 |
| A2 | 1.72 | 1.39 | 6.02 | 55.4 | 21.1 | 654 | 38.1 |
| AB | 1.39 | 1.42 | 5.88 | 55.9 | 21.9 | 649 | 38.4 |
| B1 | 0.90 | 1.58 | 5.11 | 50.3 | 24.2 | 627 | 34.5 |
| B2 | 0.59 | 1.57 | 5.27 | 40.1 | 23.6 | 628 | 27.5 |
| C | 0.57 | 1.57 | 5.28 | 30.4 | 24.0 | 628 | 20.9 |

The change in height of the Schottky anomaly at $T_1$ represents the most prominent difference between samples. As reflected in the evolution of the parameter $a_1$ and the entropy release $\Delta S_1$, the size of the anomaly and therefore the number of two-level centers per formula unit decreases with increasing iron content. This decrease contradicts the expectation that an increase of the iron content leads to an increase of the density of magnetic impurities and hence two-level centers. Instead, the opposite evolution appears to take place in the specific heat of the samples of FeSi in our study. Such a counter-intuitive behavior, in combination with the field dependence of the maximum, which is suggestive of a magnetic origin, indicates that the Schottky anomaly may not be readily connected with a two-level energy scheme arising from single iron impurities only. Adding a further aspect, the characteristic temperature $T_1$ remains essentially unchanged under increasing iron content, i.e., it does not appear to scale in an obvious way with the initial starting composition. This behavior contrasts the characteristic temperature of the onset of the saturation of the resistivity in regime IV [19, 20, 25].

In view of the sample dependence of the specific heat reported in this study, the lack of an anomaly in specific heat measurements carried out on thin needles grown from tin flux reported in Ref. [24] suggests substantial iron excess. Similarly, as discussed above, the relatively small anomaly reported for samples grown from vapor transport is consistent with iron excess, reflected also in the magnetization and Hall effect [16]. Taken together, these results motivate further studies on the interplay of impurities with the bulk and transport properties in FeSi, made possible by the optical floating-zone technique and the precise control of the starting compositions associated with it [23].

## 4 Conclusions

The specific heat of the correlated small-gap semiconductor FeSi was studied for a series of single crystals prepared from slightly different starting compositions [19, 20]. All samples studied exhibit a shallow maximum in $C/T$ between 2 K and 10 K, reminiscent of a Schottky anomaly. Under magnetic field, this anomaly decreases in size, suggestive of a magnetic origin. However, as a function of increasing initial iron content, implying an increase of the density of magnetic impurities as observed in the magnetization, the height of the anomaly decreases. Further studies are needed to clarify if and in which way the specific heat at low temperature may be related to the robust high-mobility surface conduction channel [19, 20, 24].

# Acknowledgements

We wish to thank A. Engelhardt and S. Mayr for fruitful discussions and assistance with the experiments. We also wish to thank the anonymous referee for pointing out the importance of the specific heat contribution of Apiezon N grease for our analysis.

**Funding information**    This study has been funded by the Deutsche Forschungsgemeinschaft (DFG, German Research Foundation) under TRR80 (From Electronic Correlations to Functionality, Project No. 107745057, Project E1), SPP2137 (Skyrmionics, Project No. 403191981, Grant PF393/19), and the excellence cluster MCQST under Germany's Excellence Strategy EXC-2111 (Project No. 390814868). Financial support by the European Research Council (ERC) through Advanced Grants No. 291079 (TOPFIT) and No. 788031 (ExQuiSid) is gratefully acknowledged.

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
