# Peer review of "Composition dependence of the specific heat of FeSi"

_SciPost Physics Proceedings, doi:SciPost Phys. Proc. 11, 020 (2023)_

## Round 1 · Referee Report · Anonymous (Referee 1) · 2022-9-20

Strengths

1) It's nice to see sample dependence (or Fe stoichiometry?) dependence examined.
2) This provides a way to distinguish samples in future.
3) Well written, clearly presented.

Weaknesses

1) It is not clear what underlying parameter controls gamma. In particular, the stoichiometry in the samples does not appear to have been measured or refined, and there are no rocking curves to check crystallinity.
2) There isn't really a conclusion.

Report

Related to the topic of the poster, appears to meet the criteria.

Requested changes

Apiezon N grease has a broad glass transition around 210-250 K which is not resolved with the default addenda spacing, leading to artifacts around 200-260 K in the subtracted data, but it would likely be resolved if the long-pulse approach were also used for the addenda. (Normally H grease is used above 180-200 K to avoid these artifacts.) Readers should know how the addenda were taken, and the authors should address whether the feature around 200 K could be due to the grease.

In line 40, "optically float-zoned single crystals" suggests that existing crystals were subsequently treated in a floating zone furnace, and "float-zoning" as a verb is an uncommon slang.

In line 78, the d in "d-electron" should be italicized.

There are capitalization issues in lines 140 (Si) and 172 (Hubbard).

---

## Round 2 · Author Response

We wish to thank the Referee for their effort made to review our paper. They made us aware of an important issue in our analysis that has been corrected in our revised manuscript. We hope that the referee is satisfied with the modifications and recommends the revised manuscript for publication in SciPost.

Regarding the weaknesses pointed out by the Referee, we reply: Following revision of the data analysis (see requested changes), we find no sample dependence of the linear specific heat coefficient γ within experimental accuracy. We want to point out that a quantitative analysis of the precise compositions of samples in the concentration range studied is, to the best of our knowledge, currently not possible. For instance, both energy-dispersive x-ray spectroscopy and lab-based powder x-ray diffraction did not allow us to resolve the iron concentrations in our samples with the required accuracy.

Possible insights may be accessible by virtue of the direct determination of defect concentrations, most notably vacancies and antisite disorder. For instance, we have determined the concentration of point defects in the compositional series Mn1+xSi, an isostructural sibling of FeSi, using positron annihilation spectroscopy in combination with ab-initio calculations [M. Reiner et al., Sci. Rep. 6, 29109 (2016)]. The feasibility of positron-lifetime spectroscopy has been reported on a single composition of FeSi [A. Bharathi et al., Phys. Rev. B 55, R13385 (1997)]. However, a comprehensive determination of point defect concentrations using positrons is well beyond the scope of the work reported here. We have added a corresponding discussion to the manuscript.

Regarding the requested changes by the Referee, we reply: We wish to thank the Referee for this remark. Further measurements of the heat capacity confirmed the presence of a substantial contribution of Apiezon N grease, as suggested by the Referee. In turn, we repeated all of our specific heat measurements, carefully determining the specific heat contribution of the grease prior to the measurements on each sample. We find that for all samples investigated, the specific heat tracks each other above ~20 K. At low temperatures, a shallow maximum is observed, consistent with the literature [S. Paschen et al., Phys. Rev. B 56, 12916 (1997)]. This maximum may be attributed to a Schottky anomaly, is suppressed under magnetic field, and decreases in size with increasing iron content.

In the revised manuscript, we have updated the account of the experimental methods. Moreover, we have exchanged the figures and rewritten the paragraphs describing and discussing our experimental data. As our main conclusion, we identify a magnetic field and composition dependence of the low-temperature specific heat of FeSi that contrasts the expected behavior of simple magnetic impurities.

The minor mistakes pointed out by the Referee have been corrected.

---

## Round 2 · List of Changes

As described in detail in the author comments, large parts of the manuscript were updated in response to the Referee’s remarks.

---

## Editorial Decision

published